# Testing the stability of theory of mind: A longitudinal approach

**Diane Poulin-Dubois**  *◎, **Naomi Azar**◎, **Brandon Elkaim, Kimberly Burnside**

Psychology Department, Centre for Research in Human Development, Concordia University, Montréal, Québec, Canada

◎ These authors contributed equally to this work.
* diane.poulindubois@concordia.ca

## Abstract

An explicit understanding of false belief develops around the age of four years. However, tasks based on spontaneous responses have revealed an implicit understanding of belief and other theory of mind constructs in infants in their second year of life. The few longitudinal studies that have examined conceptual continuity of theory of mind from infancy to early childhood have reported mixed findings. Here we report two longitudinal experiments to investigate the developmental relation between implicit and explicit theory of mind. No link was observed in the first experiment between false belief and intention understanding measured at 14 and 18 months with the violation of expectation paradigm and tasks measuring explicit and implicit false belief at four or five years of age. In the second experiment, infants aged 18 months were tested with a battery of tasks that measured knowledge inference and false belief. They were then tested with the theory of mind scale at five years of age. The parents completed the Children's Social Understanding Scale (CSUS) and the Social Communication Questionnaire (SCQ). As in the first experiment, there were no associations between early and later forms of theory of mind. We suggest that these findings do not support the view that there is conceptual continuity in theory of mind development.

## Introduction

Defined as the understanding that others have mental states that can differ from one's own, Theory of Mind (ToM) understanding was initially thought to emerge around four years of age [1]. However, recent studies conducted with preverbal infants and preschoolers, using tasks with minimal processing demands, have provided further insight on the development of ToM understanding [2]. A number of researchers have proposed a rich view of ToM, suggesting that it develops during the second year of life [3–9]. This view of ToM development posits that infants and younger children fail the traditional explicit ToM tasks because these tasks are heavily based on language abilities and executive functions [10]. Support for this mentalistic account comes from improved performance when task demands are reduced, such as in implicit tasks based on non-elicited or spontaneous responses [2]. These tasks are often based on looking time (violation of expectation, anticipatory looking tasks) or on spontaneous helping behaviors. In contrast to explicit false belief understanding, implicit false belief is triggered by a fast, unconscious, and efficient early developing mindreading system [11].

**Funding:** This research was supported by graduate fellowships from the Social Sciences and

Humanities Research Council of Canada (SSHRC: https://www.sshrc-crsh.gc.ca) to Kimberly Burnside (#752-2016-2436) and an Insight research grant from the Social Sciences and Humanities Research Council of Canada (#435-2017-0564) to Diane Poulin-Dubois. The funders had no role in study design, data collection and analysis, decision to publish, or preparation of the manuscript.

**Competing interests:** The authors have declared that no competing interests exist.

Other researchers have brought forward the idea that behaviors observed in implicit ToM tasks might not be based on the same knowledge as in older children and adults, but rather reflect separate ToM systems altogether, or can be explained by submentalizing processes [11]. Specifically, it has been proposed that there is an "efficient mindreading system [that] is evolutionarily and ontogenetically ancient, operates quickly, and is largely automatic and independent of central cognitive resources" (i.e., implicit ToM), and a "flexible mindreading system [that] develops late, operates slowly, and makes substantial demands on executive control processes" (i.e., explicit ToM) [12]. If this view is correct, then the efficient system should remain relatively distinct from the more flexible system [12]. There is mounting evidence that when measured concurrently, implicit and explicit false belief appear to be dissociated. Grosse Wiesmann and colleagues [13] found that, although both 3- and 4-year-olds could pass an anticipatory looking false belief task, only 4-year-olds passed the explicit location and contents false belief tasks. In a recent longitudinal study, an anticipatory looking false belief task was administered to 2-, 3-, and 4-year-olds. In this study, only at 4 years of age did children pass the anticipatory looking false belief task [14]. More importantly, no relation was observed between the differential looking score (DLS) on the anticipatory looking task and performance on explicit tasks at 4 years. Low and Watts [15] also found a dissociation between implicit and explicit location false belief in 3-year-olds. In contrast, Low [16] found that implicit false belief was a significant predictor of explicit false belief in 3- and 4-year-olds when measured concurrently. Although informative, these conflicting findings can only provide indirect information about the nature of implicit theory of mind. A more direct approach is to conduct longitudinal studies which provide a critical source of evidence in the current debate about the nature of infants' theory of mind skills. If, as it has been argued by the supporters of the mentalistic account, implicit tasks measure a construct that is mature and sophisticated but masked by task demands, then conceptual continuity should be expected from infancy to early childhood.

Some longitudinal studies have examined whether the understanding of ToM constructs in infancy predicts later explicit ToM. In a pioneering study, Wellman and colleagues [17] assessed ToM understanding in children who had participated in a study on intentional action as infants [see 18]. When they were 14 months old, infants were tested on goal attribution: they were habituated to an actor showing interest towards one of two objects, followed by a consistent test event (actor grabs the same toy) and an inconsistent test event (actor grabs another toy—not the one that she had shown interest in). At the age of 4 years, the children were administered the Wellman and Liu [19] ToM scale. Infants' decrement of attention (i.e., how fast infants habituated) during the goal attribution task was positively related to their performance on the ToM scale. It was concluded that these findings provide some evidence of stability in socio-cognitive abilities from infancy to childhood. Similarly, Aschersleben and colleagues [20] investigated whether infants' performance on a goal-directed action task, adapted from the paradigm used by Woodward [21], was correlated to their performance on the Wellman and Liu [19] ToM scale. In the goal-directed action task, infants were habituated to an actor's hand pushing an object to a designated circle. Following this, infants viewed two types of test trials: a "path change" trial and an "object change" trial. Participants' decrement of attention in the goal-directed action task was correlated with their scores on the combined false belief tasks. Yamaguchi and colleagues [22] also found stability from 12-month-olds' understanding of goal-directed actions to their ToM understanding at four years. Finally, Olineck and Poulin-Dubois [23] found that intentional action understanding in infancy, which was measured with the reenactment of the intended acts task, predicted later intention understanding, assessed with a target-hitting task at 4 years of age. Therefore, taken together, these studies demonstrate that goal and intention understanding in infancy are related to ToM constructs in childhood.

Given that the rich view of ToM is derived from infants' behaviors on implicit false belief tasks, such as the violation-of-expectation (VOE) task, there are surprisingly very few longitudinal studies that have investigated stability in the development of false belief understanding and none that have tested the stability of implicit knowledge inference or intention understanding to later belief reasoning. To date, only two longitudinal studies have tested the stability of false belief understanding from infancy to childhood [24, 25]. This is important to highlight because false belief is considered the gold standard in the assessment of ToM given that it involves the understanding that others have a different belief from one's own and that beliefs could be inconsistent with reality [26]. As such, false belief tasks measure whether individuals can truly perceive others' unobservable mental states independently from their own beliefs about a given situation [26, 27]. Thoermer and colleagues [25] found that false belief understanding at 18 months, as measured with the anticipatory looking paradigm, predicted explicit location false belief (where the child has to predict where the protagonist will search for an object that was moved in his or her absence) but not explicit contents false belief (where the child has to predict the protagonist's knowledge of the content of a box) at 4 years of age. In a more recent study, the same task administered at 18 months did not predict explicit false belief in a morally relevant context–a lack of replication explained by the task demands of the morally relevant ToM task [24]. Taken together, it appears that there is some weak stability between implicit false belief in infancy and explicit false belief in early childhood. Given the limited number of longitudinal studies testing the link between early and later theory of mind, the present study addressed this gap by examining how infants' understanding of knowledge access and goal attribution predicts later explicit and implicit false belief.

All the implicit false belief tasks used in the longitudinal studies described above are based on the anticipatory looking paradigm, where children's first look is used to assess whether they can anticipate a protagonist's actions. However, the VOE paradigm was initially used to reveal infants' false belief understanding, which involved showing infants a protagonist hiding an object in one of two boxes [6]. In her absence, the object was moved to another box. Following this change of location, the protagonist returned and either reached in the full box (i.e., incongruent with her false belief) or in the empty box (i.e., congruent with her false belief). If the infants' looking time is longer during the incongruent trials, it is assumed that they are surprised by the protagonist's action and thus, it is concluded that infants have an implicit false belief understanding. Onishi and Baillargeon [6] reported that 15-month-olds look longer during incongruent trials, thus demonstrating an implicit false belief understanding. Moreover, nonverbal interactive tasks designed to test infant's false belief construct have also revealed that by the age of 18 months, infants will attempt to help an experimenter find a toy based on her belief (true or false) about its current location [3]. In a second experiment, we examined the stability of false belief reasoning with interactive tasks, including those measuring false belief and knowledge inference.

In the first experiment, two groups of infants, aged 14 and 18 months, were tested on two VOE tasks: false belief and goal-directed action. These participants were drawn from a larger sample tested on a large range of ToM tasks [see 28]. They were re-tested at 4 or 5 years of age respectively on an anticipatory looking false belief task, a standard location false belief task, and a standard contents false belief task. The goals of this study were to 1) investigate the predictive nature of implicit ToM in infancy for implicit and explicit false belief in childhood using a longitudinal design and 2) determine if implicit and explicit false belief concepts are inter-related when measured concurrently in early childhood. Importantly, this study attempts to replicate Thoermer and colleagues' [25] longitudinal study using the widely cited false belief task originally used with infants [6]. This is an important step as it could provide unique information about the generalizability of false belief stability. Furthermore, this study also aims to

contribute to the debate about the two-systems theory [29], as both implicit and explicit false belief tasks were administered in childhood. Finally, a comparison between performance on an implicit false belief task in infancy and performance on both implicit and explicit tasks later in childhood provides both a conservative (implicit to explicit) and lenient (implicit to implicit) test of the stability hypothesis. Given previous results, we expected to find a dissociation between implicit and explicit false belief in childhood. More importantly, it was expected that implicit false belief in infancy, when measured using the VOE paradigm, would predict later implicit false belief in childhood. Finally, in line with Thoermer and colleagues' [25] findings with regards to task specificity, it was expected that the VOE false belief scores would predict explicit location false belief, but not explicit contents false belief. A goal-directed action task was included because past research demonstrated that infants' decrement of attention on goal-directed action tasks was positively related to later performance on the ToM scale or false belief tasks [17, 20, 22].

## Study 1

### Method

**Participants (Wave 1).**  Participants are a subsample of infants that participated in a larger study on theory of mind in infancy [28]. The subsample consisted of a group of twenty-five 14-month-olds ($M_{age}$ = 14.5 months, range = 13.6–15.3 months, 16 males) and a group of forty 18-month-olds ($M_{age}$ = 18.6 months, range = 17.4–20.4 months, 20 males). Families were recruited using birth lists provided by a government health agency in a large Canadian city. Infants were tested in their mother tongue, either English ($n$ = 38) or French ($n$ = 27). The study was approved by the Human Research Ethics Committee of Concordia University. (Certification # 10000548). Written consent was obtained by the parent or legal guardian of the children participants

**Participants (Wave 2).**  The 14-month-old group was re-tested at the age of 4 years ($M_{age}$ = 4.1 years, range = 4.0–4.25 years) and the 18-month-old group was re-tested at the age of 5 years ($M_{age}$ = 5.9 years, range = 5.6–6.2 years). The age gap was due to the fact that, in Wave 1, the 18-month-olds were tested first and the 14-month-olds about 12–24 months later whereas both groups were tested within a 6-month period at Wave 2. All participants were typically developing. One child was diagnosed with Autism Spectrum Disorder between Waves 1 and 2 and was therefore excluded from the final sample at both waves.

**Procedure and materials (Wave 1).**  Infants were tested in two 45-minute visits taking place one week apart. For each visit, families received a $20 compensation and infants were given a gift and a certificate of merit. The congruent trials and incongruent trials of the VOE tasks were administered during separate experimental sessions. The order of the tasks and trials within each session was counterbalanced. No order effects were observed. Infants' looking time was coded using INTERACT 8.0 (Mangold, 2010). A separate experimenter, blind to the hypotheses of the study, coded 25% of the videos. Pearson product-moment correlations revealed high inter-rater agreement ($r$ > .90) for both VOE tasks.

*VOE false belief task*. This task was adapted from Onishi and Baillargeon [6]. The task was administered on a stage-like apparatus, where a yellow and a green box (14 cm x 14 cm x 14 cm) were placed 18 cm apart. Each box had a 14 cm x 14 cm opening on the side that was covered with fabric. A red cup (7.5 cm x 10.5 cm height) or a yellow duck (11cm x 11 cm) was placed between both boxes.

Infants were seated in a highchair or on their parent's lap at about 110 cm from the stage. They viewed three familiarization trials, an induction trial, and a test trial for both conditions (congruent and incongruent), administered on separate days, one week apart. The order in

which they viewed the two conditions (first day vs. second day) was counterbalanced. Each trial was followed by an infant-directed pause, which ended if the infant looked away from the scene for more than two consecutive seconds after looking at the scene for a minimum of two cumulative seconds or looked away from the scene for 10 consecutive seconds. A trial lasted a maximum of 30 seconds.

For the first familiarization trial, the experimenter placed the toy inside one of the two boxes and paused with her hand inside the box. The cup remained in place for the next familiarization trials. In the following familiarization trials, the experimenter placed her hand inside the same box and paused in this position. In the induction trial, the toy was moved to the other box using a magnet beneath the stage while the experimenter was absent. In the test trials the experimenter either reached inside the box containing the cup (incongruent with her false belief) or inside the empty box (congruent with her false belief). The green box was always located on the right and the yellow box on the left and the object moved from right to left, as in the FB-green condition in Onishi and Baillargeon [6]. In the original study by Onishi and Baillargeon [6], there was also a FB-yellow condition where the object moved from left to right. Importantly, there were no differences between those two conditions in their sample [6].

The infants' looking time to the scene during the infant-directed pauses was calculated. A proportion score (i.e., ranging from 0 to 1) was computed by dividing the looking time during the incongruent trial by the total looking time during the congruent and incongruent trial. Participants were excluded if they failed to complete both test trials of the false belief task ($n$ = 17), yielding a final sample of 47 infants.

*VOE goal-directed action*. This task was adapted from Phillips and Wellman [30]. Infants took part in three familiarization trials and four test trials. In the familiarization trials, the experimenter stood behind a black barrier and a yellow duck was placed on the table on the other side of the barrier. The experimenter reached over the barrier to grab the duck, after which, infants' looking time was calculated until they looked away from the scene for more than two consecutive seconds after looking at the scene for a minimum of two cumulative seconds or looked away from the scene for 10 consecutive seconds. A trial lasted a maximum of 30 seconds. For the test trials, the barrier was removed. Two test trials were congruent, where the experimenter reached directly for the duck, and two test trials were incongruent, where she reached for the yellow duck as if there was still the barrier in front of her. Infants' looking time during the congruent and the incongruent trials was calculated. Looking times during the pair of congruent test trials and the pair of incongruent test trials were averaged in order to obtain the average looking time during each type of test trial. Proportion of looking time was also computed for this task by dividing the average looking time during the incongruent trials by the sum of the average congruent and average incongruent looking time. Participants were excluded if they failed to complete the four test trials ($n$ = 25), yielding a final sample of 39 for the goal-directed action task.

**Procedure and materials (Wave 2).** Children returned to the laboratory for a single visit lasting approximately one hour. The order of the tasks was semi-counterbalanced, such that the tasks administered at a table were grouped in a block to avoid a room change, which could increase fussiness and fatigue. This created four different orders. At the end of the visit, families received a $30 compensation and children were given a small gift and a certificate of participation.

*Verbal ability*. Form A of the Peabody Picture Vocabulary Test, Fourth Edition (PPVT-4) [31] was administered to participants whose dominant language was English. Form A of the Échelle de vocabulaire en images Peabody (ÉVIP) [32] was administered to participants whose dominant language was French. The ÉVIP is an adaptation of the PPVT Revised (PPVT-R).

Children's verbal age equivalent and percentiles were calculated using the norms provided by the authors of each measure.

*Explicit false belief.* The same two belief tasks (contents false belief and location false belief) used by Thoermer and colleagues [25] were administered. Tasks were adapted from the Wellman and Liu [19] ToM Scale. Participants were excluded from the analyses if they completed neither task ($n = 3$), yielding a final sample of 61 for these tasks.

Children's responses were coded offline. A separate experimenter, blind to the hypotheses of the study, coded 25% of the sample. For these tasks, we observed perfect inter-rater agreement ($r = 1.0$).

The **Contents False Belief** task began as the experimenter showed the children a Band-Aid® box and asked them what they think is inside. In order for the trial to continue, they had to answer "Band-Aids®" (or an equivalent term). If they gave another answer or failed to respond, the experimenter prompted the child until the correct answer was obtained. The experimenter then asked the children to open the box to reveal that it actually contained a toy horse. The horse was then placed back in the box and the lid was closed. The children were asked to tell the experimenter what was inside the box, as a verification question. If the child correctly said horse, the experimenter moved on. If the child got the control question wrong, the child was shown the horse once more, and re-verified that the child knew there was a horse inside the box. Following this verification, a figurine of a man, named Peter, was introduced to the children who were told that "Peter never saw inside the Band-Aid® box". They were then asked "What does Peter think is in the box? Band-Aids or a horse?", and the control question "Did he ever see inside the box?". Children passed this task if they answered that Peter thinks that there are Band-Aids® in the box and that he has never seen inside the box.

For the **Location False Belief** task, the experimenter introduced a woman figurine, named Linda and told the children that Linda lost her cat. While showing two colourful illustrations (10 cm x 6.5 cm) of trees and of a garage, the experimenter explained that "the cat might be hiding in the trees or in the garage" and that while "Linda thinks her cat is in [one of the two locations], it is actually in the [other location]". The actual location of the cat was counterbalanced across participants. The children were then asked "Where will Linda look for her cat? In the trees or in the garage?" and the control question "Where is her cat actually hiding?". They passed if they were able to correctly identify that Linda will look for her cat in the location she believes it to be and that the cat is actually in the opposite location.

*Anticipatory looking implicit false belief.* This task, also known as the "autobox task", was adapted from Thoermer and colleagues [25]. A Tobii TX300 eye-tracker (Tobii Technology, Stockholm, Sweden) was used to record children's looking time. The children were seated 60–70 cm away from a 23-inch, screen (30˚ visual angle) with a 1920 x 1080 pixel resolution. The accuracy of the Tobii TX300 is 0.4–1˚ when viewing the screen at a distance of 65cm (equivalent to an average on-screen error of 12-30mm). Sampling rate of the data was 120 Hz. The areas of interest (the two doors) were 355 x 325 pixels in width and height.

Following a 5-point calibration, the children viewed two familiarization trials, lasting 32 seconds each, and one test trial, lasting 41 seconds. In the familiarization trials, the protagonist, located at the top of the screen, watched as a yellow car moved from one garage on one side of the screen to another garage on the other side of the screen. One familiarization trial showed the car moving from left to right and the other trial from right to left (counterbalanced). Once the car reached the second garage, the protagonist disappeared, and two doors located above each garage became bright red, signalling an anticipatory looking period lasting 3 seconds. Following this, the protagonist opened the door above the second garage and grabbed the car. During the test trial, before the car reached the second garage, a phone ring distracted the protagonist, who failed to see the car reverse and exit the scene. As such, this implicit false belief

task is considered a low demand task because there is less "response-inhibition" involved (i.e., children do not have the impulse to look where the object is hiding) [33]. This was followed by a three-second anticipatory looking period, after which the protagonist opened the door where she had last seen the car. Total looking time to both doors during the anticipatory looking periods were recorded by the eye-tracker. Looks shorter than 80ms were excluded from the analyses. A differential looking score (DLS) was calculated by subtracting looking time to the incorrect door from the looking time to the correct door and dividing this difference by the total looking time to both doors during the anticipatory looking period [34–36]. A positive DLS indicated that the children looked longer at the correct door during the anticipatory looking period. Participants were excluded from the DLS analyses if they did not look at the screen during the test trial ($n$ = 1) or if they failed both familiarization trials ($n$ = 18), yielding a final sample of 45 for the DLS analyses for the implicit false belief task. Additionally, children's first look during the anticipatory looking period was coded. Participants were excluded from the first look analyses if they failed both familiarization trials (n = 25), yielding a final sample of 38 for the first look analyses of the implicit false belief task.

## Results and discussion

**Data cleaning.** Using a z-score cut-off of ± 3.0, two outliers in the VOE false belief task and one outlier for the PPVT verbal age equivalent were replaced with the closest value within normal range [37]. The other tasks, as well as chronological age, contained no outliers and were normally distributed. The false discovery rate procedure [38] was applied to all correlations run in this study in order to control for multiple comparisons.

**Wave 1.** In the original study [28], an effect of trial was reported at the trend level for the false belief task but no main effect of age-group nor any interaction effect. On the goal-directed action task, there was a main effect of trial, but no main effect of age-group nor any interaction effect. Because a subsample was tested at Wave 2, we report this group's results on the tasks measured at Wave 1. Given that a within-subjects design was used, infants viewed the congruent and incongruent trials of both VOE false belief task one week apart (counterbalanced). As such, analyses were conducted to determine if there was an order effect for both VOE tasks. Infants' proportion of looking time did not differ across orders for the false belief task ($t(42)$ = -1.29, $p$ = .20, $d$ =.-21). As for the VOE goal-directed action task, there was no difference in the scores based on whether the task was done at the first or second visit ($t(35)$ = -.95, $p$ = .35, $d$ =.-33) and whether the congruent or incongruent trial was done first ($t(36)$ = -1.13, $p$ = .27, $d$ = -.17).

*VOE false belief.* An a priori power analysis determined a required sample size of 44 participants to achieve a moderate effect size ($d$ = .50) and strong power (1 –β = .90). The 14-month-olds looked longer at the incongruent trials, as demonstrated by above chance-level proportion of looking time ($M$ = .61, $SD$ = .18, range = .17 –.85; $t(13)$ = 2.25, $p$ = .04, $d$ = .60, 95% CI [-.00, .21]). The 18-month-olds looked equally on both trials ($M$ = .53, $SD$ = .19, range = .17 –.86, $t(29)$ = .81, $p$ = .43, $d$ = .15, 95% CI [-.04, .10]). The 18-month-olds and the 14-month-olds did not differ in their performance on the VOE false belief task ($t(42)$ = -1.30, $p$ = .20, $d$ = -.22, 95% CI [-.20, .04]). Given the lack of group differences, to gain maximum statistical power, data from both age groups was combined in the analyses using the looking time proportion scores. Infants' average proportion score on the VOE false belief task was not statistically different from chance ($M$ = .55, $SD$ = .19, range = .17 –.86, $t(43)$ = 1.87, $p$ = .07, $d$ = .28, 95% CI [-.00, .11]). Additionally, infants' performance on the VOE false belief task was not correlated with age ($r(42)$ = -.23, $p$ = .13).

*VOE goal-directed action.* An a priori power analysis determined a required sample size of 44 participants to achieve a moderate effect size ($d$ = .50) and strong power (1 –β = .90). The

14-month-olds looked equally on both trials ($M$ = .52, $SD$ = .12, range = .33 –.74, $t(15)$ = .56, $p$ = .58, $d$ = .14), whereas the 18-month-olds looked longer in the incongruent trial than in the congruent trial ($M$ = .60, $SD$ = .14, range = .36 –.82, $t(21)$ = 3.31, $p$ = .003, $d$ = .70). The 18-month-olds and the 14-month-olds did not differ in their performance on the VOE goal-directed action task ($t(36)$ = 1.91, $p$ = .06, $d$ = .24). Infants' average proportion score was statistically different from chance ($M$ = .56, $SD$ = .14, range = .33 –.82, $t(37)$ = 2.91, $p$ = .01, $d$ = .47). Infants' performance on the task was not correlated with age ($r(36)$ = .31, $p$ = .06). Finally, infants' proportion score on the VOE false belief task was not correlated with their proportion score on the VOE goal-directed action task ($r(36)$ = -.03, $p$ = .85).

**Wave 2.** *Verbal ability*. Children's verbal age equivalent, as measured using the PPVT or the ÉVIP was on average 5.82 years ($SD$ = 1.63).

*Anticipatory looking implicit false belief.* An a priori power analysis determined a required sample size of 44 participants to achieve a moderate effect size ($d$ = .50) and strong power (1 – β = .90). For the DLS analyses, whether or not children started or ended the visit with the implicit task had no effect on performance ($t(43)$ = 1.14, $p$ = .26). Forty-five (72%) children passed at least one familiarization trial (i.e., looked longer at the correct door). Children's DLS was not statistically different from chance ($M$ = -.08, $SD$ = .66, $t(44)$ = -.77, $p$ = .44, $d$ = -.11). Furthermore, the DLS was not correlated with chronological age ($r(45)$ = -.06, $p$ = .72). Specifically, the 4-year-olds' average DLS was -.09 ($SD$ = .64) and the 5-year-olds' average DLS was -.05 ($SD$ = .72). Moreover, the DLS was not correlated with verbal ability (see Table 1).

For the first look analyses, whether or not children started or ended the visit with the implicit task had no effect on performance ($\chi^2(1)$ = .03, $p$ = .86). Thirty-eight (60%) children passed at least one familiarization trial, that is, first look towards the door from which the protagonist appeared. Of these, two participants were inattentive at test. Only 22% (8/36) of the children's first look was directed to the correct door with a passing rate statistically below chance (binomial $p$ = .001). Children's performance on this task was not correlated with age ($r(34)$ = -.05, $p$ = .77). Specifically, 23% (3/13) of the 4-year-olds, and 22% (5/23) of the 5-year-olds, directed their first look to the correct door. Children's first look was also not correlated with verbal ability. Lastly, performance as measured by the DLS was not correlated with performance as measured by first look.

**Table 1. Zero-order correlations between measures at Wave 2 in Experiment 1.**

| | Implicit FB DLS | Implicit FB First Look | Location FB | Contents FB | Explicit FB Score | Verbal Ability |
|---|---|---|---|---|---|---|
| **Implicit FB DLS** | 1 | $r_{pb}$ = .07 | $r_{pb}$ = -.23 | $r_{pb}$ = -.02 | $r$ = -.15 | $r$ = -.13 |
| | | N = 36 | N = 44 | N = 44 | N = 44 | N = 44 |
| **Implicit FB First Look** | | 1 | φ = .21 | = -.03 | $r_{pb}$ = .11 | $r_{pb}$ = -.05 |
| | | | N = 34 | N = 34 | N = 34 | N = 34 |
| **Location FB** | | | 1 | φ = .28* | $r_{pb}$ = .79** | $r_{pb}$ = .26 |
| | | | | N = 61 | N = 61 | N = 58 |
| **Contents FB** | | | | 1 | $r_{pb}$ = .81** | $r_{pb}$ = .54** |
| | | | | | N = 61 | N = 58 |
| **Explicit FB Score** | | | | | 1 | $r$ = .50** |
| | | | | | | N = 58 |
| **Verbal Ability** | | | | | | 1 |

*Notes.*

\* $p < .05$

\*\* $p < .01$. *Abbreviations*. FB: False Belief; DLS: Differential Looking Score. *Symbols*. r: Pearson correlation; $r_{pb}$: Point-biserial Pearson correlation; φ: Phi coefficient.

*Explicit false belief.* There was no difference in performance on the explicit composite score between children who completed the implicit task first or last ($t(59) = .40$, $p = .692$). The order in which the two explicit tasks were done did not impact children's composite score ($t(59) = 1.17$, $p = .245$) or their performance on the contents false belief task ($\chi^2(1) = .14$, $p = .705$). However, children who completed the location false belief task first were more likely to fail that task compared to those who started with the contents task ($\chi^2(1) = 5.44$, $p = .020$).

A proportion of 34% (21/61) of the children passed the location false belief task, which was statistically below chance (binomial $p = .02$). Children's performance on this task was not correlated with age ($r_{pb}(61) = .17$, $p = .20$), with 26% (6/23) of the children in the younger age group and 39% (15/38) of the children in the older age group passed the location false belief task. Children's performance on this task was not significantly correlated with their verbal ability. Conversely, 48% (29/61) passed the explicit contents false belief task, a proportion not different than chance (binomial $p = .80$). Children's performance on the contents task was statistically correlated with age ($r_{pb}(61) = .57$, $p < .001$). Specifically, 13% (3/23, binomial $p < .001$) of the children in the younger age group and 68% (26/38, binomial $p = .034$) of the children in the older age group passed the task. Children's performance on this task was also positively correlated with their verbal ability. Since the two explicit false belief tasks were coded categorically (i.e., pass/fail), an explicit false belief composite score was computed (scores ranged from 0 to 2). Children's composite score was at chance level ($M = .82$, $SD = .79$, $t(60) = -1.79$, $p = .08$, $d = -.23$). Furthermore, their composite score was statistically correlated to their age ($r(61) = .47$, $p < .001$). As with the individual tasks, children's composite score was also positively correlated with their verbal ability.

Children's DLS on the implicit false belief task was not correlated with their performance on the explicit location task, the explicit contents task, nor with their composite score, indicating a dissociation between implicit and explicit false belief in early childhood. When controlling for age, children's DLS on the implicit false belief task was still unrelated to their explicit composite score ($r(41) = -.15$, $p = .34$). Children's first look during the implicit false belief task was not related to their performance on the location task, to their performance on the contents task, nor to their composite score. When controlling for age, children's first look during the implicit false belief task was still unrelated to their explicit composite score ($r_{pb}(31) = .15$, $p = .40$). However, their performance on the explicit tasks was positively correlated.

**Cross-wave analyses.** An a priori power analysis determined a required sample size of 47 participants to achieve a moderate effect size ($r = .45$) and strong power ($1 - \beta = .90$). Infants' performance on both VOE tasks was not correlated with any of their scores at Wave 2, for both implicit and explicit tasks (see Table 2).

*Additional analyses.* Given that most longitudinal studies on theory of mind have reported that habituation rate, but not test trials, predicts later ToM, we calculated a decrement of attention score on the familiarization trials of the VOE false belief and VOE goal-directed action

**Table 2. Zero-order correlations between measures at Wave 1 and 2 in Experiment 1.**

| | | Wave 2 | | | | |
|---|---|---|---|---|---|---|
| | | **Implicit FB DLS** | **Implicit FB First Look** | **Location FB** | **Contents FB** | **Explicit FB Score** |
| **Wave 1** | **VOE FB Proportion Score** | $r = .31$ | $r_{pb} = .12$ | $r_{pb} = -.02$ | $r_{pb} = -.25$ | $r = -.18$ |
| | | N = 34 | N = 27 | N = 42 | N = 42 | N = 42 |
| | **VOE GD Proportion Score** | $r = .06$ | $r_{pb} = .21$ | $r_{pb} = .10$ | $r_{pb} = .10$ | $r = .12$ |
| | | N = 27 | N = 23 | N = 35 | N = 35 | N = 35 |

*Abbreviations.* FB: False Belief; DLS: Differential Looking Score. *Symbols.* r: Pearson correlation; $r_{pb}$: Point-biserial Pearson correlation.

tasks and examined its relation to later performance in childhood. The decrement of attention on the VOE goal-directed action task ($M = 4.33$, $SD = 7.71$) and on the VOE false belief task (Incongruent trials: $M = 3.88$, $SD = 7.14$; Congruent trials: $M = 8.33$, $SD = 8.59$) did not predict any of the childhood scores.

## Discussion

The main goals of this first longitudinal study were to determine 1) whether implicit and explicit false belief tasks are dissociated in early childhood as predicted by the two-systems theory, and 2) whether performance in false belief and/or goal attribution tasks in infancy predicts either implicit or explicit false belief understanding in early childhood. Although children performed poorly on some of the tasks, there was enough individual variability in each task to conduct a test of stability. The main findings were as follows: First, as reported in previous research, a dissociation was observed across the implicit and explicit tasks in preschoolers. Second, and also in line with previous research, language skills were related to performance on the explicit but not implicit false belief tasks. Finally, and more importantly, no stability was observed from infancy to childhood on theory of mind constructs, including false belief and object-directed action processing. Given the observed chance performance at the group level on both the VOE false belief task in infancy and the anticipatory looking task in childhood, caution is advised in interpreting the present null results. It is worth noting that since the present study was completed, other researchers have also failed to replicate the VOE false belief task [39–41] while others have been successful, despite some methodological changes [42]. One reason for the present lack of replication of the VOE FB task might be that a within-subject design was used instead of the classic between-subject design. Despite the counterbalancing of the congruent and incongruent condition and the one-week delay between conditions, negative findings can be obtained in VOE tasks when there is repeated testing with perceptually similar events [43]. Furthermore, the anticipatory looking paradigm has proven challenging to replicate since the present study was completed [44]. In contrast to false belief, the other theory of mind construct measured in infancy (i.e., the goal-directedness of *successful* intentional actions) was observed to be robust despite being measured with a within-subject design. This confirms that this ability develops by the end of the first year and before implicit false belief, when measured with the same VOE paradigm [28, 45]. Nevertheless, there was also no link between performance on this task and later theory of mind tasks. This suggests that it is possible for infants to appreciate such actions as goal-directed without necessarily appreciating intentions [46, 47]. Thus, potentially, an infant could identify the goal-state and goal-object that an actor is moving toward, without identifying the mental states that guide the actor (i.e., the actor wants the object). Our findings do not replicate previous research showing longitudinal links between behavioral and neural markers of action processing and explicit theory of mind (ToM) [17, 20, 22, 24, 25, 48–51]. Thus, it remains unclear why a link emerges in some measures of action encoding and not others.

In order to replicate and expand the results of the first study, a second longitudinal study was conducted with a different battery of theory of mind tasks in both infancy and early childhood.

## Study 2

Given that children's performance on some tasks was not significantly different from chance and that the associations between false belief in infancy and childhood were null in our first study, we aimed to test the longitudinal (dis)continuity of ToM using an additional ToM construct (i.e., knowledge inference) and an interaction-based false belief task in infancy in a

second, independent sample. We also measured explicit theory of mind more broadly with the Wellman and Liu [19] ToM scale as well as with parent report questionnaires in childhood. Recently, Tahiroglu and colleagues [52] created the Children's Social Understanding Scale (CSUS) to assess theory of mind through a parent-report questionnaire. The CSUS can replace traditional standardized testing, accounting for parents' perceptions of their children's social understanding. Using a wide range of explicit tasks from the Wellman and Liu [19] scale as well as an indirect measure of theory of mind based on parental report provides a way of exploring more extensively the impact of task modalities on theory of mind stability as well the breadth of such stability. In the case of broad stability, one would expect that infants' constructs will predict global measures of later theory of mind. In contrast, limited, task-specific stability would be reflected in task-specific stability (e.g., implicit knowledge inference and explicit knowledge access).

## Method

**Participants (Wave 1).**   Participants belong to a sample ($N = 66$, 35 females) of infants who participated in a larger study [53]. Infants were tested at 18-months of age ($M_{age} = 18.52$ months, range = 17.40–20.00 months). Families were recruited using birth lists provided by a government health agency in a large Canadian city. Infants were screened for perinatal risk, developmental disabilities, or audio-visual impairments. Participants were tested in their mother tongue, which was either French ($n = 29$) or English ($n = 37$). The study was approved by the Human Research Ethics Committee of Concordia University. (Certification # 10000548). Written consent was obtained by the parent or legal guardian of the children participant.

**Participants (Wave 2).**   Children returned to the laboratory at the age of five years ($M_{age} = 55.9$ months, range = 52–61 months). Participants were screened on the phone for developmental disabilities or audio-visual impairments prior to the study visit. During this screening, two participants were identified as meeting the exclusion criteria since Wave 1. Therefore, those two participants did not participate in the study at Wave 2 and their data were not included in any of the analyses that follow. Participants were tested in their mother tongue, which was either French ($n = 28$) or English ($n = 38$). Ethical approval to conduct this study was granted by the institutional review board of the University.

**Procedure and materials (Wave 1).**   Infants were tested during a single one-hour visit. Families received a $20 compensation and infants were given a gift and a certificate of merit.

*False belief ToM task*. An interactive false belief task was used to examine infants' understanding that others may have false beliefs [3]. In the original task by Buttelmann and colleagues [3], infants were seated on the floor, one meter away from the two boxes. In order to minimize the original high attrition rate (43% of the original sample did not help or helped with parental assistance) the task was modified by having the infant sit at a table. This way, infants did not need to move toward the boxes and there was no need to change the set-up between the tasks. The task started with one experimenter (E1) announcing that she was leaving the room to get a toy. During E1's departure, a second experimenter (E2) familiarized the child with a set of two 30 x 30 x 30 cm boxes with wooden pins, which were placed on the end of a table, out of reach from the infant. E2 then showed the infant how to lock the toy boxes. E1 returned and placed her toy in one of the boxes as the infant was watching. She then announced that she was leaving again to go find her keys. While E1 was out of the room, E2 invited the infant to play a trick on E1 and move the toy to the opposite box. When E1 returned, she tried to open the box where she had left her toy and displayed confusion and disappointment when she was not able to open the box and retrieve her toy. E2 then pushed the

boxes toward the infant and E1 prompted the infant to help her find the toy. The correct answer on this task required the child to consider that E1 was not in the room when the toy was moved and that she therefore held a false belief about the toy's location. The trial was coded as pass/fail, where a pass was given to the infant for choosing the box that currently contained the toy. Such a response demonstrates an understanding of E1's false belief. The colour (green or orange) and location (left or right) of the box in which E1 first places the toy was counterbalanced, creating four orders. A Cohen's kappa coefficient was computed and indicated perfect inter-rater agreement ($\kappa = 1.0$).

*Knowledge inference task.* Another ToM task was used to assess infants' knowledge inference abilities [54]. In this task, infants need to demonstrate an understanding that others may have access to different knowledge than them and therefore, make different inferences. This task begins with a familiarization trial where the two experimenters and the infant play with three familiar objects (a ball, a car, and a teddy bear) for 50 seconds. Next, in a pre-test trial, E1 asked for each toy one by one in order to verify the infant's ability to share with the experimenter. To pass this pre-test, infants needed to have given the experimenter at least one of the first two objects requested. E1 then announced that she was going to play at the other end of the room. E2 retrieved a novel toy (a plastic gardening tool) and brought it to E1 to play with for 30 seconds while the infant was watching. Next, E2 retrieved the toy and allowed the infant to play with it for 30 seconds. This procedure was repeated to introduce a second novel object (a modified bird-cage mirror). After playing with each toy, E2 placed it on a tray on the table. E1 then announced that she is leaving the room and, while she is away, E2 introduced a third novel object (a small modified abacus). When E1 returned, she exclaimed "Oh, look! Look there! Look at that there! Can you give it to me please?", while pointing at the tray. In order to pass this task, the infant is required to give E1 the novel toy introduced while she was out of the room. This response reflects an understanding of E1's reaction toward the third toy and that E1 does not have knowledge about it. The order in which the toys were introduced and their placement on the tray were counterbalanced, creating nine orders. A Cohen's kappa coefficient was computed and indicated excellent inter-rater agreement ($\kappa = .88$).

**Procedure and materials (Wave 2).** Children returned to the laboratory for a single visit lasting approximately one hour. The primary caregiver completed the Children's Social Understanding Scale (CSUS) and the Social Communication Questionnaire (SCQ) during the testing session. At the end of the visit, families received a $30 compensation and children were given a small gift and a certificate of merit.

The testing session always began with the administration of the Wellman and Liu [19] ToM Scale. The Diverse Desires, Diverse Beliefs, Knowledge Access, Content False Belief, and Hidden Emotions tasks were used, resulting in a five-item scale. Diverse Desires was always administered first as it is the easiest item while Hidden Emotions was always administered last as it is the hardest item. The order in which Diverse Beliefs, Knowledge Access, and Contents False Belief were administered was counterbalanced, resulting in six orders (3x2x1). All participants completed the entire scale, except for one child who was excluded on the Contents False Belief task due to difficulty staying on task and following instructions. Following the completion of the Wellman and Liu [19] ToM Scale, the participants completed either the Peabody Picture Vocabulary Test (PPVT) or the Échelle de vocabulaire en images Peabody (ÉVIP), depending on their dominant language.

*ToM Scale.* Children's responses were coded both live and offline. All tasks were double coded, and double entered for 100% of cases. For all tasks, we observed perfect inter-rater agreement ($r = 1.0$).

The experimenter began the **Diverse Desires** task by introducing the children to a figurine of an elderly man named Mr. Jones, who was hungry and wanted a snack to eat. They were

then shown two snack options that Mr. Jones could eat: carrots or cookies. The order of presentation of snack options was counterbalanced. Mr. Jones was placed in between both snack options, after which the children were asked to choose the snack that they would prefer to eat. Upon their own selection, Mr. Jones was placed in front of the opposite snack. The children were told "Well [child's snack preference] are a really good choice! But Mr. Jones likes [opposite snack]. He doesn't like [child's snack preference]. What he likes best are [opposite snack]". Immediately after, children were reminded that Mr. Jones is hungry and that he could only choose one snack to eat. They were then prompted to select the snack that Mr. Jones would like best. A pass on this task requires the children to pick the opposite snack for Mr. Jones, relative to their own preference.

Next, the **Diverse Desires** task is similar to the Location False Belief task used in Study 1 but formulated to represent diverse beliefs rather than false beliefs. As in Study 1, the experimenter introduced the figurine of Linda, who was looking to find her cat, and the two possible two hiding locations that Linda could look in: the garage or the bushes. The order of presentation of hiding options was counterbalanced. Linda was placed in between both hiding options. The children were then asked where they thought the cat was hiding. Following their answer, Linda was placed in front of the opposite location. The children were told "Well that's a really good idea! But Linda thinks her cat is hiding in the [opposite location]. Linda thinks her cat is in the [opposite location]". The children were then prompted to select the location where Linda would look for her cat. To pass this task, children were required to pick the location opposite to their own belief.

For the **Knowledge Access** task, the experimenter placed a gift-style box on the table. The experimenter asked the children to guess what they thought was inside the box. Following their guess, or if the children did not want to guess, the experimenter opened the box to reveal a figurine dog inside the box and said: "Wow! Look! It's a dog inside the box". Following the reveal, the dog was place back inside the box, and the box was closed. The children were then asked to tell the experimenter what was inside the box, as a control question. If the child correctly said dog, the experimenter moved on. If the child got the control question wrong, the child was shown the dog once more, and re-verified that the child knew there was a dog inside the box. Following verification of the control question, Polly, a teenage figurine, was introduced and placed on the table in front of the box. The children were told that Polly had never, ever seen inside the box. The experimenter then brought Polly next to the box and asked the children "Now, here comes Polly. So, does Polly know what is inside the box? Has she ever seen inside what is inside the box, and that she had never seen inside the box. Correctly answering only one of two questions was treated as a failure of the item.

The **Contents False Belief** task is similar to the one administered in Study 1 with some methodological differences in order to follow the exact script in the original Wellman and Liu [19] validation paper. The experimenter began by presenting a Band-Aid® box and asked them what they thought was inside. The children could make any response before continuing. The rest of the task was administered exactly as in Study 1.

The last task was the **Hidden Emotion** task. The experimenter first showed the children a sheet presenting three smiley faces. The experimenter verified that the children could correctly identify the "happy", "okay", and "sad" face. Next, the children were given instructions about the task. They were told that they were going to hear a story about a boy and be asked (a) "how he really feels inside", and (b) "how he looks on his face". The children were told that the boy "might really feel one way inside but look a different way on his face. Or, he might really feel the same way inside as he looks on his face". The experimenter then told the story of a boy named Matt. Matt was playing with his friends when one of the other children, Rosie, told a mean joke about him to the others. Everyone was laughing, but Matt did not think it was

funny. Matt did not want others to see how he felt because they would "call him a baby". For the duration of the story, a cartoon of a boy was shown to them, although the face of the boy was not visible. Following the story, two control questions were asked. First, the children were asked what the other children would do when Rosie told a mean joke about Matt. They were then asked what the other children would do if they knew how Matt really felt. Following the control questions, the children were asked to identify how Matt really felt inside, and how he looked on his face using the previously trained smiley faces. In order to pass this item, the child had to select a less severe smiley face for how Matt tried to look on his face, as compared to how Matt felt inside. For example, if Matt really felt sad inside, he had to look "okay" or "happy" on his face.

*Verbal ability*. As in Study 1, the PPVT-4 [31] was administered to participants whose dominant language was English while the ÉVIP [32] was administered whose dominant language was French.

*Children's Social Understanding Scale (CSUS)*. The Children's Social Understanding Scale [52] or its French adaptation [55] was administered to parents of participants to obtain a parental report of their children's ToM abilities. Parents rated their children's ToM abilities on a 42-item rating scale. A total score as well as 6 subscale scores were calculated, namely Belief, Intent, Emotion, Knowledge, Perception, and Desire. Higher scores on this measure indicate greater social understanding and ToM abilities.

*Social Communication Questionnaire (SCQ)*. The Social Communication Questionnaire [56] was administered to parents of participants to obtain a parental report of their children's communication skills and social functions. This questionnaire provides one total score, based on 40 items answered as yes or no. Higher scores on this measure indicate greater dysfunction in social communication abilities. A Quebec French adaptation was created for the French-speaking participants.

## Results

**Data cleaning.**   Using a z-score cut-off of ±3.0, one outlier on each of the Intent and Knowledge subscales of the CSUS were replaced with the closest value within normal range [37]. All other variables were normally distributed and did not contain any outliers. The false discovery rate procedure [38] was applied to all correlations run in this study in order to control for multiple comparisons.

**Wave 1.**   In the original study that included a larger sample, infants' performance on the false belief ToM task was not significantly different from chance while performance on the knowledge ToM task (touching the target) was significantly above chance. Because a subsample was tested at Wave 2, we report the subgroup's results on the tasks measured at Wave 1.

*False belief ToM task*. Three participants were excluded due to fussiness or inattentiveness. Descriptive statistics indicated that, 50.8% of infants touched the correct box. A binomial test indicated that this proportion was not statistically different from chance (.50) ($p$ = 1.000). These results are consistent with those reported on the larger original sample [53]. There were no order effects based on the colour or location of the box in which E1 first placed the toy ($\chi^2(3)$ = 2.91, p = .405).

*Knowledge ToM task*. Thirteen participants were excluded on this task due to fussiness, not touching any of the target objects, or failing the pretest task. Descriptive statistics indicated that 43.4% of infants touched the correct object. A binomial test indicated that this proportion was significantly above chance (.33) ($p$ < .001). These results are consistent with those of the original study as well as with our previous findings based on the full sample [53, 54].

**Wave 2.** *Verbal ability.* Children's verbal age equivalent, as measured using the PPVT or the ÉVIP was on average 4.72 years (SD = 1.33). Following correction for multiple comparisons, scores were not significantly related to age (r(64) = .25, p = .048).

*ToM scale.* The total scale score as well as individual subscale scores were considered in order to provide broader, more detailed information. The Diverse Desires task was passed by 66.7% of children. A binomial test indicated this proportion to be significantly above chance (.50) (*p* = .009). The Diverse Beliefs task was passed by 71.2% of children, which was also significantly above chance (*p* = .001). As for Knowledge Access, it was passed by 48.5% of children, which is not significantly different from chance (*p* = .902). The Content False Belief task was passed by only 15.4% of children, which was significantly below chance (*p* < .001). Importantly, when asked to guess what was inside the Band-Aid® box, only 30.3% (n = 20) correctly guessed that it was Band-Aids® while all children recognized that the box actually contained a horse. Lastly, 40.9% of children passed the Hidden Emotion task, which was not significantly different from chance (*p* = .175). The mean score on the total scale was 2.44 (*SD* = 1.22). There was no difference on total score across the six different task orders (*F*(6, 59) = .49, *p* = .816). As expected, age was significantly correlated with the total Scale score (*r*(64) = .38, *p* = .002) after correction.

*CSUS.* A proportion score was computed for this measure because one of the answer options on the questionnaire is "don't know". As such, the sum of scores of completed item scores was divided by the number of items answered, excluding those answered with "don't know". The mean total score was 3.33 (*SD* = .34, range = 2.27–3.98). After correction for multiple comparisons, the total score on the CSUS was not correlated with age or verbal ability. In addition, none of the CSUS subscales or its total score were correlated with any of the explicit ToM Scale scores or its total score.

*SCQ.* The mean score on the SCQ was 6.48 (*SD* = 4.18, range = 0.00–17.00). After correction for multiple comparisons, scores on this measure were not correlated with age or with verbal ability. In addition, they were not correlated with performance on the global ToM Scale or any of its individual tasks. Following correction, SCQ scores were significantly correlated only with the CSUS Perception subscale (*r*(64) = -.36, *p* = .003).

**Cross-wave analyses.** After correction for multiple comparisons, infants' performance on both ToM tasks at Wave 1 was not correlated with any of their scores at Wave 2 (see Table 3).

## Discussion

In conclusion, the results from this second experiment replicate the null results observed in the first experiment. None of the expected associations between early and later forms of ToM were observed. More specifically, infants who performed better on the false belief task at the age of 18 months did not perform better on a wide battery of tasks measuring theory of mind in childhood. Interestingly, this contrasts with the previously reported conceptual continuity between implicit false belief in infancy based on the anticipatory looking procedure and an explicit location false belief [25]. As is the case with past findings, no such link was observed with an explicit content false belief task, but the poor performance on this task could explain such null findings.

The interactive false belief task that we used in the present study has produced highly variable results across studies [44] and has not been replicated in a number of recent studies [41, 53, 57]. It has recently been concluded that there is currently no robust evidence from interaction tasks for early belief ascription. Based on the findings from strict and conceptual replication attempts. Nevertheless, we agree with the conclusion that by proactively helping the adult who was not present when the object was moved, children showed that they knew that visual

**Table 3. Zero-order correlations between tasks at Wave 1 and Wave 2 in Experiment 2.**

| | | Wave 1 FB | Wave 1 KN |
|---|---|---|---|
| Wave 2 ToM subscale and composite scores | Diverse Desires | $\phi$ = -.09 | $\phi$ = -.14 |
| | | n = 63 | n = 53 |
| | Diverse Beliefs | $\phi$ = .01 | $\phi$ = -.04 |
| | | n = 63 | n = 53 |
| | Knowledge Access | $\phi$ = -.08 | $\phi$ = .05 |
| | | n = 63 | n = 53 |
| | Content False Belief | $\phi$ = -.24 | $\phi$ = .07 |
| | | n = 62 | n = 53 |
| | Hidden Emotions | $\phi$ = .02 | $\phi$ = .04 |
| | | n = 63 | n = 53 |
| | Composite score | $r_{pb}$ = -.10 | $r_{pb}$ = .03 |
| | | n = 63 | n = 53 |
| Wave 2 CSUS | Total score | $r_{pb}$ = .02 | $r_{pb}$ = -.17 |
| | | n = 63 | n = 53 |
| | Belief | $r_{pb}$ = .12 | $r_{pb}$ = -.17 |
| | | n = 63 | n = 53 |
| | Knowledge | $r_{pb}$ = .08 | $r_{pb}$ = -.13 |
| | | n = 63 | n = 53 |
| | Perception | $r_{pb}$ = -.08 | $r_{pb}$ = -.10 |
| | | n = 63 | n = 53 |
| | Desire | $r_{pb}$ = .02 | $r_{pb}$ = -.21 |
| | | n = 63 | n = 53 |
| | Intention | $r_{pb}$ = .06 | $r_{pb}$ = -.12 |
| | | n = 63 | n = 53 |
| | Emotion | $r_{pb}$ = -.07 | $r_{pb}$ = -.02 |
| | | n = 63 | n = 53 |
| Wave 2 SCQ | Total score | $r_{pb}$ = .18 | $r_{pb}$ = -.01 |
| | | n = 63 | n = 53 |

*Abbreviations*. FB: False Belief; KN: Knowledge; ToM: Theory of Mind; CSUS: Children's Social Understanding Scale; SCQ: Social Communication Questionnaire. *Symbols*. $r_{pb}$: Point-biserial Pearson correlation; $\phi$: Phi coefficient.

access is necessary to succeed on this hide-and-seek game [58]. This ability could certainly provide a building block or even a precursor to full-fledged false belief reasoning. As for the knowledge inference task, although we successfully replicated the original findings, no longitudinal link was observed with later explicit tasks measuring knowledge access, either directly in laboratory- based task or indirectly with parental report. Given that it is the first time that such continuity is tested, future research should examine such link with other tasks measuring infants' understanding that others might possess knowledge that differ from their own [59].

## General discussion

The main objective of the current set of studies was to assess the stability of false belief understanding from infancy to early childhood. To date, only three published studies have examined the stability of false belief from infancy to early childhood in the same German sample [24, 25, 60]. The longitudinal design for the current studies was closely modeled after that of Thoermer and colleagues [25] who found that infants' false belief tracking, assessed with an anticipatory looking task at 18 months, significantly predicted their false belief understanding at 4 years of

age. More recently, the same team reported that the anticipatory looking task at 18 months was significantly associated with performance on both the explicit location and contents false belief tasks at 50, 60, and 70 months of age [60]. In the first experiment, we had hypothesized that stability would be similarly observed when the VOE task is substituted to the anticipatory looking task, given that both tasks are based on spontaneous responses. It was also deemed critical to test false belief stability with both implicit and explicit false belief tasks in childhood as it would allow both a conservative (VOE vs. standard false belief tasks) and a lenient (VOE vs. anticipatory looking tasks) test of the stability hypothesis. The results failed to show the conceptual stability previously reported in the German sample followed longitudinally between 18 and 70 months [25, 60]. This was reflected in a lack of statistically significant correlations between infants' performance on the VOE task at 14 and 18 months and their performance at 4 or 5 years on both implicit and explicit false belief tasks, as well as in the null results observed in the regression models. Importantly, Sodian and colleagues [24] found that false belief understanding at 18 months of age, measured with the same anticipatory looking task, was not a predictor of later false belief understanding, which was assessed using a moral interview task. Furthermore, infants' performance on the goal-directed action task (i.e., proportion of looking time and decrement of attention) was not predictive of later false belief understanding. Other researchers have been successful in showing stability of basic ToM skills from infancy to childhood. For example, 14-month-olds' rate of habituation during the familiarization trials of a task measuring intentional action was found to be related to their ToM understanding at 5 years of age [17]. However, it is important to note that only one study has found such a link when the test trials were used as the predictor [24]. This pattern of results has been replicated in other longitudinal studies, in which the researchers were able to show that infants' processing of goal-directed actions (i.e., decrement of attention) was correlated with scores on the ToM scale when assessed in childhood [20, 22]. To address the limitations of the first experiment, we conducted a second longitudinal experiment with a new sample of children tested on a set of theory of mind tasks that were not based on looking time measures. In line with the first experiment, we observed no significant association.

In sum, the present findings provide no evidence of stability from socio-cognitive skills in infancy to belief understanding during the preschool years. Such lack of conceptual continuity from infancy to preschool age is inconsistent with a rich interpretation of infants' behaviors in theory of mind tasks. If a mature false belief understanding emerges in infancy but is masked by task demands, then such competence should be stable over time. We did not observe such stability across a number of constructs, including goal-directed action, knowledge access, and false belief. Indeed, support for the mentalistic account of infant ToM would require observing a relation across conceptually equivalent tasks at each developmental period. However, as pointed out by Thoermer and colleagues (25), to systematically test whether continuity is task-specific requires a design that includes, for instance, both location and content false belief in infancy as well as in preschool age. It is therefore not possible to test for task-specific relations between an implicit version of a content false-belief task in infancy and the content task in preschool age. Importantly, the present findings indicate no specific relation between implicit and explicit location false belief tasks, nor between implicit and explicit knowledge access tasks, when the tasks varied in whether action prediction (explicit) or action interpretation (implicit) were required. The fact that we tested infants' and preschoolers' theory of mind with different paradigms that vary in task demands and consistently found null results suggests that the observed discontinuity is robust. It is worth noting that a lack of continuity was observed across tasks with both low (VOE false belief anticipatory looking) and high (VOE goal-directed action, knowledge access) success rates. Moreover, the replication of the null effects with a

measurement of theory of mind that has no task demands (i.e., parental report) provides additional validity to our conclusions.

The lack of evidence for stability that we observed in the first experiment might indicate that the false belief task tested with the VOE paradigm does not assess false belief but other abilities, as suggested by many lean interpretations of the behaviors observed in this task [61]. We concur with the view that success on implicit false belief tasks in both infants and apes is probably not achieved by submentalizing, but rather reflects the ability to track what the agent sees and has seen and how this will affect the agent's behavior [62, 63]. In other words, there is no need to understand that the agent's belief diverges from one's own belief to perform well on infant false belief tasks. According to Tomasello's [62] shared intentionality account, engaging in social and mental coordination with others, along with skills of executive function, is necessary for children to become able to coordinate their own perspective with both the other person's perspective and the objective situation. Support for this position comes from the strong links reported between infants' joint attention skills and explicit ToM in childhood [64–66]. This suggests that some building blocks are laid down in infancy for the development of later ToM understanding, but that the VOE false belief task, as currently used in infancy, might not be the best measure of such abilities. The fact that infants attribute false beliefs to an inanimate object or generalize beliefs to ignorant agents suggests that this task measures, at best, an immature form of theory of mind [67, 68]. Future research will be required to establish the construct validity for implicit false belief tasks, an endeavor currently undertaken by the ManyBabies2 project, which aims to conduct strict replications of the anticipatory looking, VOE, and interactive tasks in a large number of laboratories [69].

Regarding the anticipatory looking task that we used, preschool children performed at chance (i.e., looked at both doors equally) as did the 18-month-olds in the original longitudinal study by Thoermer and colleagues [25]. This is also in line with some recent failed attempts to replicate this specific task (the "autobox" task) in both children and adults [35, 70]. Moreover, several failed attempts at replicating other versions of the anticipatory looking paradigm have recently been reported in both adults and children [13, 14, 41, 71, 72]. Nevertheless, the variability in performance on this task allowed us to determine whether implicit false belief was related to both implicit false belief in infants and concurrent explicit false belief. Interestingly, and in accord with previous research, verbal skills did not relate to performance on this task in contrast to the well-established link between explicit false belief tasks and language [73]. Furthermore, performance on implicit and explicit theory of mind tasks in childhood did not converge, suggesting that distinct cognitive processes underlie implicit and explicit false belief reasoning, as recently confirmed in a study showing dissociated brain regions involved in processing implicit and explicit false belief tasks [74]. The results are therefore compatible with a dual process view of implicit and explicit ToM which suggests an automatic, cognitively efficient (possibly unconscious) belief-tracking system already present in infancy, and an explicit more flexible but cognitively more demanding belief processing system, which develops later [12, 29, 75, 76].

The present set of studies was designed with the hope of providing additional information to the current debate about the depth of infants' ToM development. Stability of false belief understanding in a longitudinal study could provide further evidence for the rich view of ToM development, whereas a lack of evidence of stability would support the lean interpretation of ToM tasks. Results from both experiments fail to provide support for stability, while also challenging the view that precursors to theory of mind in the form of preconceptual socio-cognitive abilities provide the foundation for the development of theory of mind skills [11]. Other researchers have suggested that infant false belief measures tap into implicit false belief

attribution processes which then evolve into an explicit understanding through developing language, social interactions, and executive control abilities [e.g., 60, 77, 78].

Of course, absence of evidence does not entail evidence of absence so we suggest that future research should focus on strict replications of past studies on infants' false belief understanding as well as short-term longitudinal designs, such as testing false belief with the VOE paradigm at 18 months and the anticipatory looking paradigm a few months later, or vice-versa. This would constitute evidence for stability in implicit false belief tracking based on spontaneous responses. Furthermore, given that infants' performance on the VOE paradigm, in contrast to the anticipatory looking paradigm [25], does not predict later explicit false belief understanding, it is of importance to establish convergent validity by using different paradigms (e.g., VOE and anticipatory looking) concurrently in infancy to determine if these tasks are capturing the same construct. Assessing the stability of false belief understanding using a battery of tasks is a necessity to determine if different tasks are better able to capture this construct [79]. Furthermore, test-retest reliability of the same task would further help clarify whether false belief understanding is stable throughout development. The full pattern of continuities from early socio-cognitive skills to later ToM is an important topic for future research.

## Supporting information

**S1 Data.**
(XLSX)

## Acknowledgments

The authors would like to thank Mallorie Brisson and Jean-Louis René for their help with data collection and coding. The authors would also like to thank all the families who kindly participated in this study throughout the years.

## Author Contributions

**Conceptualization:** Diane Poulin-Dubois, Naomi Azar, Kimberly Burnside.

**Data curation:** Naomi Azar, Kimberly Burnside.

**Formal analysis:** Naomi Azar, Kimberly Burnside.

**Funding acquisition:** Diane Poulin-Dubois.

**Investigation:** Naomi Azar, Brandon Elkaim, Kimberly Burnside.

**Methodology:** Diane Poulin-Dubois, Naomi Azar, Brandon Elkaim, Kimberly Burnside.

**Project administration:** Diane Poulin-Dubois.

**Resources:** Diane Poulin-Dubois.

**Supervision:** Diane Poulin-Dubois.

**Visualization:** Naomi Azar.

**Writing – original draft:** Diane Poulin-Dubois, Naomi Azar, Kimberly Burnside.

**Writing – review & editing:** Diane Poulin-Dubois, Naomi Azar, Brandon Elkaim.

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
