## [Decision Letter · Decision Letter 0]

27 Jul 2020

PONE-D-20-12030

Testing the stability of theory of mind: A longitudinal approach

PLOS ONE

Dear Dr. Poulin-Dubois,

Thank you for submitting your manuscript to PLOS ONE. After careful consideration, we feel that it has merit but does not fully meet PLOS ONE’s publication criteria as it currently stands. Therefore, we invite you to submit a revised version of the manuscript that addresses the points raised during the review process.

I agree with the two reviewers that a major revision is needed. The reviewers provide detailed comments of what needs to be clarified, explained, and modified. Please address all comments of the reviewers in your revised manuscript. Some main points, in brief: please provide more clear theoretical motivation for the study and the specific items used from the Wellman & Liu (2004) scale. Clarify how different accounts interpret and predict performance; what differentiates implicit and explicit theory of mind tasks; statistical tests should be modified and also streamline the results; the conclusions should be stated more tentatively, given many non-significant findings; expand discussion of the results of all ToM tasks and clarify the seemingly contradictory statements. 

We look forward to receiving your revised manuscript.

Kind regards,

Julie Jeannette Gros-Louis, PhD

Academic Editor

PLOS ONE

Journal Requirements:

2. We note that some details in the Methods and Results section have been removed for blinding purposes. Please ensure that all citations are updated and details included in your revised manuscript. Please be aware that PLOS ONE operates a single-blind peer review process, where the identity of authors is known to editors and reviewers.

3. We note you have included a table to which you do not refer in the text of your manuscript. Please ensure that you refer to Table 7 in your text; if accepted, production will need this reference to link the reader to the Table.

Reviewers' comments:

Reviewer's Responses to Questions

**Comments to the Author**

1. Is the manuscript technically sound, and do the data support the conclusions?

Reviewer #1: Partly

Reviewer #2: Partly

2. Has the statistical analysis been performed appropriately and rigorously? 

Reviewer #1: Yes

Reviewer #2: I Don't Know

3. Have the authors made all data underlying the findings in their manuscript fully available?

Reviewer #1: Yes

Reviewer #2: Yes

4. Is the manuscript presented in an intelligible fashion and written in standard English?

Reviewer #1: Yes

Reviewer #2: Yes

5. Review Comments to the Author

Reviewer #1: The goal of the current study was to examine whether there is continuity between infants’ performance on implicit Theory of Mind (ToM) measures and their later performance on ToM tasks during the preschool years. In study 1, children were tested using false-belief and intention understanding Violation of Expectation (VOE) paradigms in infancy and traditional false-belief tasks at ages 4 and 5. There were no associations between children’s performance on the tasks in infancy and their later performance at 4 and 5. In study 2, 18-month-olds completed false-belief and knowledge inference tasks using interactive paradigms. These children were tested again at age 5 using the Wellman and Liu (2004) Theory of Mind scale and several other measures. There was an association between infants false-belief performance and their later performance on the divergent beliefs measure from the Wellman and Liu scale at age 5. No other associations were reported.

The question of whether early performance on ToM tasks is predictive of children’s later ToM understanding is an important one that is likely to be of interest to the field. However, there are several issues with the manuscript that need to be addressed.

The introduction is difficult to follow and the theoretical motivation for the study is not clearly or strongly outlined. For instance, the authors describe competing views regarding the development of ToM, but they do not explain how these accounts interpret infants’ performance on implicit ToM tasks, or what these different accounts would predict in terms of continuity between ToM understanding from infancy to later childhood. There is a brief mention of the “stability hypothesis” and conservative and lenient tests of this hypothesis, but this is not unpacked for the reader. In addition, hypotheses are made regarding stability of false-belief understanding across time, but what about the other aspects of ToM the authors are examining (e.g. intention understanding, knowledge inference)?

One of the key goals of the study is to examine associations between early performance on implicit and later performance on implicit and explicit ToM measures. However, these terms are never described in terms of what kinds of tasks fall under these categories. What differentiates the two kinds of tasks?

Study 2 in particular does not include much motivation for the measures used to examine associations in ToM from infancy to the preschool years. The authors used two interactive ToM tasks for infants that are very different from the looking time measures used in prior studies they are aiming to replicate and those in Study 1. There is no description of why parental report measures were included in the task battery at age 5 and how those might compare to more traditional ToM tasks in terms of their relation to infants’ ToM. Finally, there is no justification made for why individual items/measures were considered from the Wellman & Liu (2004) scale rather than considering children’s total scores or false-belief only scores as has been done in prior studies (Aschersleben et al., 2008, Wellman et al., 2004, 2008, Yamaguchi et al., 2009 ). Were there specific hypotheses regarding how the infancy tasks might related to specific items on the scale?

Study 2 involved an interactive false-belief task that has been used in prior research (Buttelmann et al., 2009). As the authors note, there have been both successful and failed attempts to replicate this task. However, there are several factors that when changed from the original task are thought to contribute chance performance among infants in this paradigm (Buttelmann et al., 2018), such as whether the child is seated on the floor vs. at a table (resulting in differences in amount of time available to process events), manipulations to the boxes that make them more salient, etc. It would be useful to know whether this study was a strict replication or whether any adaptations to the task were made in the current study.

The results sections contain redundant information. For instance, in Study 1 correlations are reported both in the text and in tables and it is not clear why additional simple regression analyses were performed (line 387), that showed very similar results to the null findings in the zero-order correlation table. Sections could be streamlined to reduce redundancy. Some of the tables (e.g., Tables 1 and 2) are somewhat cluttered. It might be useful to remove the row of p-values and instead mark significant results with asterisks or other symbols. Finally, many tests were reported in the results of this study (especially in Study 2) without giving any apriori predictions about relations between specific tasks.

In the discussion section, the authors posit that the reason they may not have found associations to later ToM with the VOE false-belief task is because it doesn’t actually measure understanding of false beliefs. However, they also state that the interactive helping task, where they did find associations to later ToM also doesn’t measure false-belief understanding. Shouldn’t it be the case then, that neither task would predict later ToM performance? There is also no interpretation and discussion of findings regarding the other ToM tasks in this section.

Other Comments

1. The authors state that the main goal of the article was to assess the stability of false-belief understanding from infancy to early childhood. However, their study involved many ToM measures other than false-belief understanding.

2. In the introduction and elsewhere there are several field-specific terms that are used that some readers may be unfamiliar with (e.g. contents, location, etc.). Brief descriptions of what those terms refer to would be helpful.

3. In sections on data cleaning, the authors note that data for outliers was replaced with the closest value within normal range. Is this a common method for replacing looking-time data and if so, can a reference for this method be provided? Even if it is a common practice, it seems odd to replace data in a study examining individual differences.

4. In the Study 1 false-belief task and goal-directed action task, please include how many seconds infants were allowed to look at the familiarization and test trials before the trial ended.

5. In the Study 1 goal-directed action task, infants saw two test trials of each kind of event (congruent, incongruent). It would be useful to have more details regarding how the proportion scores were calculated in the context of this multi-test trial task. In line with this issue, the description of the results for the goal-directed action task only mention 1 test trial. Which test trials were included in those analyses?

6. In both Study 1 and Study 2, participants were excluded from the final sample at Wave 2 due to having a developmental disability or audio-visual impairments. Were their data also removed from Wave 1?

7. In the results section, additional information regarding potential effects or interactions should be reported. For instance, for the VOE false-belief task in Study 1, were there any effects of box color, side of the box, etc.? This information should be included for any of the tasks in infancy that counterbalanced various aspects of the events and that could impact infants’ looking/performance.

8. In the description of the contents false-belief tasks it states that children had to say that they thought BandAids (or an equivalent term) were in the box before moving on with the experiment. However, in the discussion of Study 2, the fact that a large percentage of children couldn’t identify the contents of the BandAids box is given as a potential reason for why there were no associations with that task. This information seems to be in conflict.

9. The authors mention that the rich interpretation focuses on task demands playing an important role in whether children can demonstrate false-belief understanding. Given that this study involved different kinds of tasks with varying demands, it would be interesting to acknowledge this further in the discussion and reflect on how those task demands may have played a role in the current study.

Reviewer #2: Testing the stability of theory of mind: A longitudinal approach

Poulin-Dubois et al.

The manuscript presents two longitudinal studies with the aim of relating implicit Theory of Mind (ToM) tasks at the age of 14/18 months (wave 1) with implicit as well as explicit ToM tasks at the age of 4/5 years (wave 2). Study 1 assessed a violation of expectation (VOE) implicit ToM task (Onishi & Baillargeon, 2005) at wave 1 and an anticipatory looking (AL) implicit plus two traditional explicit ToM tasks (Thoermer et al. 2012) at wave 2. Study 2 assessed an interactive helping ToM task (Buttelmann et al. 2009) at wave 1 and the explicit Wellman & Liu ToM scale (Wellman & Liu 2004) at wave 2. The authors found that:

(1) In study 1, most of the task performances were at chance level, except for the explicit FB location task which was below chance.

(2) There were no correlations between tasks of wave 1 and wave 2 of study 1.

(3) In study 2, task performance on the interactive helping paradigm at 18 months (wave 1) was at chance level, and only some of the explicit ToM tasks in preschool age (wave 2) had above chance performance.

(4) Most of the tasks showed no correlations between wave 1 and wave 2 in study 2, except for the interactive task at wave 1 with the diverse beliefs tasks at wave 2.

The authors interpret these results in a way that precursors of ToM might already be present as building blocks in infancy.

These two studies are definitely of big interest for the current debate about the interpretation of infant ToM findings and the question of their continuity to later ToM development in preschool-age. Despite the apparent value of these studies for the field, we have a few comments that we think should be addressed to ensure that the reported findings and their interpretation are fully valid.

Major Comments

(1) First of all, we were wondering whether the correlations reported in study 2, including the only significant correlation in the paper, were also corrected for multiple comparison. The authors only reported correction for study 1. Since a relatively large number of correlations were tested and only a single positive correlation was reported (with p = .039), this would be essential for the finding and conclusions of the study to hold.

(2) Given a single significant amongst mostly non-significant correlations between early infant ToM task and later explicit ToM tasks we feel that the author’s conclusion that the infant findings are a ToM precursor that provides the building blocks for later mature ToM is too strong. Even if this correlation survives correction for multiple comparison, we therefore suggest to phrase the conclusions more tentatively.

(3) Another problem in the interpretation of the positive correlation as well as the null-findings is that the performance of many tasks (including the interactive helping task that shows a positive correlation with later diverse belief reasoning) is at chance-level. This makes it hard to assess whether the correlated variance in these tasks is cognitively meaningful. While we consider it important to report these null findings, especially in light of the current debate on ToM in infancy, and appreciate the authors' discussion of these null-findings with reference to future research like the ManyBabies project, the issue of the interpretation of correlations between chance-level task performances could be discussed in some more depth in the manuscript.

(4) Relatedly, the chance-level performance in some of the tasks could be discussed in some more depth – are there any deviations from the original tasks that might explain the non-replication? How do the findings in specific tasks relate to findings in other non-replications? For example, in the interactive helping task, several studies have replicated a significant difference between the false belief condition and the true belief control condition, while not replicating the above-chance choice of box in the false belief condition itself. The authors’ findings seem to be in line with these previous findings.

(5) A related issue is the interpretability of null-findings: From the lack of correlations in study 1 and part of study 2, the authors conclude a lack of stability of the tasks that may support a two system model of implicit and explicit ToM. Concerning these null-findings, it is important to note that absence of evidence is not evidence of absence. We suggest to rephrase these parts as “lack of evidence for stability/continuity” etc. rather than “lack of stability” etc. and refer to positive evidence for a dissociation as for example found in a recent paper by Grosse Wiesmann and colleagues (2020, PNAS).

(6) Comments 3 and 5 also make the conclusion of task specificity of the implicit-explicit correlation problematic as this argument builds on the presence of certain correlations but absence of others. We therefore suggest to remove or rephrase this argument.

(7) The authors included power analysis for both studies, however the assumed effect size for all tasks of 0.5 seems rather large. Is this effect size based on prior literature?

Minor Comments

(8) We suggest to include some more motivation of the task design in the introduction: why were completely different tasks used in study 2 than 1 and why were exactly these tasks chosen? Why were children in study 1 retested at different time points (4 and 5 years)?

(9) Some of the tables lack a description including the abbreviations used in the table as well as the correlation method used.

(10) It wasn’t fully clear to us whether the samples of study 1 and 2 were independent samples or whether the same children were tested with the different tasks of study 1 and 2. We suggest the authors specify this explicitly in the manuscript.

(11) References:

l. 804: We suggest to also add reference to a similar account by Southgate (2020, Psychol Review, preprint available since 2018).

l. 824: "several failed attempts at replicating other versions of the anticipatory looking paradigm have recently been reported in both adults and children (39, 58-60).” Reference 59 does not report a failed replication of anticipatory looking tasks. Instead references 13 and 14 of the same authors do.

6. PLOS authors have the option to publish the peer review history of their article (what does this mean?). If published, this will include your full peer review and any attached files.

Reviewer #1: No

Reviewer #2: No

---

## [Author Response · Author response to Decision Letter 0]

2 Oct 2020

Detailed responses are attached in a separate document

---

## [Editor Report · Decision Letter 1]

20 Oct 2020

Testing the stability of theory of mind: A longitudinal approach

PONE-D-20-12030R1

Dear Dr. Poulin-Dubois,

We’re pleased to inform you that your manuscript has been judged scientifically suitable for publication and will be formally accepted for publication once it meets all outstanding technical requirements.

Kind regards,

Julie Jeannette Gros-Louis, PhD

Academic Editor

PLOS ONE
---

## [Editor Report · Acceptance letter]

26 Oct 2020

PONE-D-20-12030R1 

Testing the stability of theory of mind: A longitudinal approach 

Dear Dr. Poulin-Dubois:

I'm pleased to inform you that your manuscript has been deemed suitable for publication in PLOS ONE. Congratulations! Your manuscript is now with our production department. 

Kind regards, 

on behalf of

Dr. Julie Jeannette Gros-Louis 

Academic Editor

PLOS ONE